# Effects of Steam Treatment Time and Drying Temperature on Properties of Sweet Basil’s Antioxidants, Aroma Compounds, Color, and Tissue Structure

**DOI:** 10.3390/foods12081663

**Published:** 2023-04-16

**Authors:** Yoko Tsurunaga, Mina Kanou

**Affiliations:** 1Faculty of Human Science, Shimane University, Shimane 690-8504, Japan; 2Graduate School of Human and Social Sciences, Shimane University, Shimane 690-8504, Japan

**Keywords:** Genova, aroma compounds, antioxidant activity, tea production

## Abstract

This study has developed a production method for high-quality Genova tea with excellent antioxidant properties. The antioxidant properties of each part of the Genova basil plant (i.e., leaves, flowers, and stems) were determined; the leaves and flowers showed higher antioxidant values. We also investigated the effects of steaming time and drying temperature on the antioxidant composition and properties, color, and aroma using leaves with good yield potential and high antioxidant properties. The color showed excellent green color retention with freeze- and machine-drying at 40 °C without steam-heat treatment. Steaming for 2 min was effective in maintaining high values of total polyphenol content, antioxidant properties (1,1-diphenyl-2-picrylhydrazine and hydrophilic oxygen radical adsorption capacity), rosmarinic acid, and chicoric acid, and a drying temperature of ≤40 °C was recommended. Freeze-drying without steaming was the best method to retain all three of Genova’s main aroma components, Linalool, *trans*-alpha-bergamotene, and 2-methoxy-3-(2-propenyl)-phenol. The method developed in this study can improve the quality of dried Genova products and be applied in the food industry, cosmetics, and pharmaceutical industries.

## 1. Introduction

Genova (*Ocimum basilicum* cv. ‘Genovese’), a cultivar of sweet basil (*Ocimum basilicum* Linn.), is a member of the Perilla family, commonly known as common basil [1]. Sweet basil is a perennial herb from the Lamiaceae family produced primarily in the regions of Genova, Imperia, and Savona in Liguria, Italy [2]. Sweet basil is used in both Ayurvedic and Unani medicine as a medicinal herb [3] and possesses antibacterial properties as well as inhibitory properties against HIV-1 reverse transcriptase and platelet aggregation [1,4,5,6]. Furthermore, aromatic compounds extracted from sweet basil are used in cosmetics and natural fragrances [3] and drunk as basil tea [1]. Genova is popular for its use in pasta sauces [7] and contains high levels of polyphenol components, such as rosmarinic acid, a caffeic acid derivative, chicoric acid, and caffeic acid, with particularly high rosmarinic acid content [8,9]. Rosmarinic acid exhibits antioxidant, anti-inflammatory, and neuroprotective properties [9] and is abundant in sweet basil and other plants of the Lamiaceae family, such as egoma and lemon balm [10,11]. Genova contains high levels of aromatic phenylpropanoids, sesquiterpenes, and monoterpenes [12]. Although Genova has the highest antioxidant potential of any sweet basil [9,13], only a few studies have been conducted on processing methods that retain its antioxidant properties. For example, there exist studies on the effects of different drying methods (microwave, sun-drying at 12.2–21.0 °C and 37–98% humidity, and machine-drying (MD) at 45, 50, or 55 °C) on ascorbic acid, phenol, and chlorophyll contents and antioxidant activity [14]; and optimization of osmotic dehydration to improve dried product quality and to limit changes in appearance, texture, flavor, and color [15]. We have previously examined the tea-making process of *Ocimum gratissimum* with respect to the steaming duration (1 or 2 min), drying method (freeze-drying (FD), MD, or shade-drying (SD)), drying temperature (40 or 80 °C), roasting (with or without), and fermentation treatment (with or without); however, detailed studies on its steaming duration and drying temperature were not available due to a lack of processing conditions [16]. If a method is established to retain more efficiently the antioxidant activity, functional components, and aroma compounds of raw materials, highly functional tea will be developed and used in cosmetics, natural fragrances, and the pharmaceutical industry. It has also been postulated that the use of Genova will expand.

Therefore, the objective of this study is to develop a strategy to process Genova for high functionality. Specifically, we determined detailed conditions for the steaming duration and drying temperature, which have not been previously reported, and elucidated the effects of tea processing on the antioxidant components and properties of Genova tea. Moreover, the color and aroma components, which are important indicators of tea quality, were also evaluated. Scanning electron microscopy (SEM) was used to verify the effects of tea processing on leaf tissue structure and, ultimately, antioxidant properties and quality.

## 2. Materials and Methods

### 2.1. Materials for Experiments

Folin–Ciocalteu reagent solution (2 N), 1,1-diphenyl-2-picrylhydrazine (DPPH, 95%) (powder), Trolox (97%), 2,2′-azobis (2-amidinopropane) dihydrochloride (AAPH, 95%) (powder), and ethanol solution (99.5%) were purchased from Wako Chemicals Ltd. (Osaka, Japan), and fluorescein sodium salt (1 mg/mL in pure water) from Sigma–Aldrich Chemie GmbH (Steinheim, Germany). Catechin (CTN), rosmarinic acid, and chicoric acid (>98%) in powdered form were purchased from Funakoshi Corporation (Tokyo, Japan).

### 2.2. Materials for Genova Tea

Samples were collected in August 2021 in Higashi-Hiroshima City, Hiroshima Prefecture, Japan (Figure 1A). The collected Genova plants were divided into three parts: leaves, flowers, and stems, to determine their total polyphenol content (TPC) and antioxidant properties with respect to the collection site.

### 2.3. Genova Tea Manufacturing Method

Only the leaves were used in the tea production test. Figure 1 presents the digital images of the Genova tea produced following each treatment. Notably, the environment in which the images were taken was identical, but the weights of the tea leaves were not. Steaming(S)-treatment was divided into six treatments: non-steaming (NS), and steaming for 1 min (S1), 2 min (S2), 3 min (S3), 5 min (S5), and 10 min (S10). A food steamer (Ultra-compact, T-Fal, Paris, France) was used for S-treatment, and FD was performed using an Alpha1-2LDplus freeze dryer (Alpha1-2LDplus, Martin Christ, Osterode, Germany) to determine only the effect of S-treatment time. The drying temperatures were 40, 50, 60, 70, 80, 90, and 100 °C using MD. All treatments were set to NS to determine only the effect of drying temperature. The time required for the leaves to dry was determined to be 2400, 420, 210, 60, 45, 26, and 22 min at 40, 50, 60, 70, 80, 90, and 100 °C, respectively. Different drying methods were also employed for comparison, namely FD (–56 °C, 48 h) and SD (25 °C, 7 days). Each treatment zone was produced in triplicate, and the samples were mixed to make them homogeneous, with one sample per treatment. After tea preparation for each treatment zone, the tea was powdered using a blender (Sunbeam Oster Inc., Boca Raton, FL, USA) [16].

### 2.4. SEM

SEM was performed to observe the changes in the leaf tissue structure due to tea processing. After processing, leaves were cut into 5 × 5 mm pieces and fixed to the SEM sample stand (Nissin EM Corporation, Type-HM, Tokyo, Japan) with double-sided carbon tape for SEM (Nissin EM Corporation, 8 mm × 20 m, Tokyo, Japan). Gold deposition was applied, and the leaf surface was observed under the SEM (JEOL, JSM-IT100, Tokyo, Japan) at an acceleration voltage of 10 kV and magnification of 250×.

### 2.5. Color

The L*, a*, and b* values of the sample powders were measured using a colorimeter (CR-13, Konica Minolta, Tokyo, Japan) according to the method described by Tsurunaga et al. [16]. L* indicates lightness, a* indicates red–green degrees, and b* indicates blue–yellow degrees. Six color measurements were taken per treatment.

### 2.6. Sample Extraction

Extraction was performed in hot water as previously described [16]. An aqueous solution (20 mL) of the powdered samples (0.05 g) was boiled for 10 min, and the final volume was adjusted to 50 mL with distilled water. The aqueous solution was used for TPC measurement, DPPH radical scavenging assay, hydrophilic oxygen radical absorbance capacity (H-ORAC) assay, and high-performance liquid chromatography (HPLC). Extraction was performed twice per treatment.

### 2.7. TPC and DPPH Radical Scavenging Assay and H-ORAC Assay

TPC and DPPH radical scavenging assays were conducted as previously described [16], and H-ORAC assay was performed as previously described [17]. TPC is mg CTN equivalent/100 g dry weight (mg CTN eq/100 g DW), and DPPH and H-ORAC values were expressed in µmol Trolox equivalent/g DW (µmol TE/g DW). Each measurement was performed in triplicate.

### 2.8. HPLC Analyses

Samples were analyzed using a quantitative HPLC system (LaChrom, Hitachi Ltd., Ibaraki, Japan), and the measurement conditions have been previously described [16]. The samples were analyzed using a quantitative HPLC system (LaChrom, Hitachi Ltd.) by comparison with standard compounds using an InertSustainSwift C18 column (4.6 × 150 mm) (GL Sciences Inc. Tokyo, Japan). The analysis conditions were as follows: UV detection, 280 nm (0–7.5 min) and 370 nm (7.5–60 min); column oven temperature, 40 °C; flow rate, 1.0 mL/min; mobile phases, (A) 0.1 % formic acid/water and (B) 0.1 % formic acid/acetonitrile; and gradient condition, 0 min → (100:0) → 2 min (90:10) → 15 min (65:35) → 20 min (65:35) → 20.10 min (5:95). For HPLC analysis, two hot water extractions were performed, and two injections were made on the same extract, for a total of four replicates of data.

### 2.9. Gas Chromatography–Mass Spectrometry

(GC-MS) analysis GC-MS-QP2020 (Shimadzu Corp., Kyoto, Japan) measurements were analyzed as previously described [16,18], which is a modification of a previous method [19]. Briefly, samples were added to a 20 mL headspace vial and warmed for 5 min at 60 °C. A solid-phase microextraction (SPME) fiber of 50/30 µm DVB/CAR/PDMS (Sigma-Aldrich, Tokyo, Japan) was inserted into the vial, and the components were extracted for 30 min at 60 °C. The SPME fiber was then inserted into a single quadrupole GC–MS system with a GCMS-QP2020 gas chromatograph and an AOC-6000 autosampler (Shimadzu, Kyoto, Japan). The analysis conditions were as follows: carrier gas, helium (constant pressure 150 kPa); row, DB-Heavy WAX 0.25 mm × 60 m, 0.25 µm (Agilent Technologies, CA, USA); vaporization chamber, split 1 min at 250 °C; temperature program, 50 °C for 4 min, with temperature then being increased by 5 °C/min increments and maintained at 250 °C for 15 min; MS conditions, scan range 30–400 *m*/*z*; ionization potential, 70 eV; ion source, 200 °C; and transfer line, 250 °C. Both extraction and injection were performed once. Compounds in the obtained particle spectra were estimated using the main EI-MS library software of the NIST/EPA/NIH Mass Spectral Library with Search Program 2017 (NIST17).

### 2.10. Statistical Analysis

SPSS Statistics was used for all statistical analysis (ver. 25, IBM Inc., Chicago, IL, USA). Multiple comparisons were performed using one-way ANOVA followed by Tukey’s test. *p* < 0.05 was considered statistically significant. Data are expressed as mean ± Standard Error (SE).

## 3. Results and Discussion

### 3.1. TPC and Antioxidant Properties according to Plant Part

The TPC and antioxidant properties according to plant part are shown in Figure 2. Leaves and flowers showed high values (Figure 2). Leaves comprise the majority of the whole plant (Figure 1A) and are most often used in tea production. In contrast, flowers can be collected only in much smaller quantities and fall off during the tea-making process, which reduces the efficiency of the process. Therefore, only leaves were used as samples to test the tea-making processing conditions.

### 3.2. SEM

SEM images for each tea processing condition are shown in Figure 1B,C. The circular entities observed in the SEM images are the glandular trichomes [20], which are frequently found in Labiatae plants and contain unique aromatic components depending on the type of plant. Previous studies have also reported observations on basil glandular trichomes [21]. The glandular trichome is easily affected by external factors, such as heat treatment, as it is an external tissue. In the present study, the tea processing resulted in structural changes, such as wrinkling in the center and sinking of the entire glandular trichome (Figure 1B,C). NS and S1 indicated residual glandular trichome shape, whereas S2 and above led to sinking (Figure 1B). In the drying method, change in the shape of the glandular trichome was not observed during FD and SD or at 40 °C and 50 °C in MD (Figure 1C). De Santana et al. [22] reported changes in the structure and essential oil content of glandular trichomes in basil (*Ocimum gratissimum* L.) with different drying methods and temperatures. The authors reported more damage to the glandular trichomes and more changes in the essential oil content at 60 °C. In this study, damage to the glandular trichome was also significantly reduced at MD ≥ 60 °C compared with that at MD 40 °C and 50 °C. The tissue structure of the whole leaf became less distinct above S10 at the steam-heat treatment time but retained its shape at 100 °C at the drying temperature, indicating that the steam-heat treatment affected its structure more significantly than the drying temperature. SEM revealed that the tissue structure of Genova leaves changes depending on the tea-making treatment. In particular, the sinking of the glandular trichome occurred above S2 for the S and above MD 60 °C for the drying method, indicating that the whole leaf tissue structure was affected more significantly by the steaming time than by the drying method.

### 3.3. Color

Leaf appearance is depicted in Figure 1, and L* (brightness), a* (red–green), and b* (blue–yellow) values are listed in Figure 3. Green fading was observed in all S upon comparing the images of NS and S for the steam heat treatment (Figure 1B). S10 showed a significantly faded green color compared with NS, S1, S2, S3, and S5 (Figure 1B). The effect of S treatment on the color was greatest for a* values (red–green), which were significantly higher for S compared with NS, indicating a more faded green color than the latter (Figure 3B). The S treatment was used for maintaining the green color by inactivating chlorophyllase, a chlorophyll-degrading enzyme. However, the results of this experiment showed that the a* values were higher in S than in NS, whereas the green color was not retained by the S treatment. Images of leaves also showed fading of the green color in S compared with that in NS (Figure 1B), which is thought to be due to the heat-induced degradation of chlorophyll to pheophytin. Chlorophyll degradation is affected by acids, temperature, enzymes, and light. Previous studies have shown that longer heating time causes more browning of green leaves [16], and the conversion of chlorophyll to pheophytin is more likely to occur under acidic conditions [23]. Phenolic acids, such as rosmarinic, chicoric, and caffeic acids, have been found in Genova [24,25]. SEM images indicated that S destroyed the leaf tissue structure compared with NS (Figure 1B). Therefore, it was inferred that the S treatment destroyed the leaf tissue structure and degraded chlorophyll into pheophytin under the influence of acidic substances, such as phenolic acids, resulting in the fading of the green color of the leaves.

We also compared the efficiency of FD and SD, which are frequently used during the drying process. MD was examined at 40, 50, 60, 70, 80, 90, and 100 °C. FD is considered the best method for maintaining quality, whereas SD is the most economical and commonly used drying method, even though it is weather-dependent [26]. FD retained the maximum green color, followed by MD 40 °C, whereas SD resulted in a faded green color (Figure 1C). This color retention effect was highest at 40 °C among the MDs, with significant green fading at > 50 °C. Significant differences in L*, a*, and b* values were observed during MD at 40 °C versus > 50 °C (Figure 3). The a* value was −4.6 ± 0.7 at 40; however, the values were positive above 50 °C: 1.0 ± 0.4 (50 °C), 3.8 ± 0.2 (60 °C), 2.0 ± 0.2 (70 °C), 2.2 ± 0.3 (80 °C), 2.4 ± 0.4 (90 °C), and 1.4 ± 0.3 (100 °C). The images also showed browning of the leaves at higher drying temperatures, consistent with the color results (Figure 1C). These results can be attributed to the heat-induced formation of a brown substance in the Maillard reaction [27]. The results indicate that drying temperature has a greater effect on the color difference of Genova tea than did S time (Figure 3). Among the L*, a*, and b* values, the a* value was the most sensitive to heat and drying temperature conditions, showing higher values with increases in heat time and drying temperature. FD showed the best results in terms of color retention. FD using sublimation is an excellent method for preserving the tea’s quality [27] but requires high capital investment and running costs. The a* value was closest to that of FD during MD 40 °C, and the color was bright, with high L* and b* values, and had an excellent appearance (Figure 1B), indicating that this drying method is effective in maintaining color at a low cost.

### 3.4. TPC and Antioxidant Activities

Figure 4 presents the results of TPC (A, a), DPPH value (B, b), and H-ORAC value (C, c). S treatment for ≥ 2 min resulted in significantly higher TPC, DPPH, and H-ORAC values compared with NS, which were 7040.2 ± 66.4 mg CTN eq/100 g DW, 508.4 ± 16.4 µmol TE/g DW, and 1448.8 ± 149.2 µmol TE/g DW, respectively. S2 values were 8319.3 ± 98.2 mg CTN eq/100 g DW, 598.2 ± 16.5 µmol TE/g DW, and 2466.6 ± 161.0 µmol TE/g DW, which were approximately 1.2, 1.2, and 1.7 times higher than NS, respectively. S1 values were significantly lower relative to NS for TPC and DPPH, and no significant difference was observed for H-ORAC. Above S2, the inactivation of polyphenol oxidase (PPO), which is a polyphenol-degrading enzyme, occurs, and a decrease in polyphenols during the lyophilization process is suppressed. It is assumed that S1 prolonged the rise in the leaf temperature, PPO inactivation was incomplete, and polyphenol content and antioxidant properties were not retained during the drying process. Among the different drying methods, FD showed the highest TPC and DPPH values (Figure 4a,b), whereas H-ORAC values were similar among FD, SD, and MD 40 °C (*p* > 0.05; Figure 4c). FD showed the best results in terms of quality preservation effects, such as composition and color [27]. However, the present study revealed that the H-ORAC activity of low-cost SD and MD 40 °C is comparable to that of FD. Although SD and MD 40 °C were minutely inferior to FD, they can maintain high values of TPC and DPPH (Figure 4a,b). During MD, all assays showed significantly lower values at 50 °C compared with that at 40 °C and a further significant decrease in activity at 60 °C compared with that at 50 °C *(p* < 0.05) (Figure 4a–c). Significant differences were not observed in any of the assays at drying temperatures > 60 °C (*p* < 0.05). PPO possibly remained active during drying because S treatment was not performed. Antioxidant properties were reduced due to oxidation reactions with PPO between 50–80 °C with respect to its active temperature range [28]. However, PPO activity is unlikely in drying treatments > 90 °C, and the decrease in its values may be primarily attributed to oxidation reactions. However, to determine this, it is necessary to measure the PPO activity value. These results indicate that S2 and drying at a temperature of ≤40 °C (FD, SD, or MD) is optimal for maintaining the antioxidant properties of Genova.

### 3.5. Rosmarinic and Chicoric Acid Content

Previous studies have reported rosmarinic and chicoric acids to be the polyphenolic components of basil [29]. Therefore, we analyzed the standards of these acids and S2, which had the highest antioxidant property, using HPLC, and detected chicoric acid at 11.2 min retention time (RT) and rosmarinic acid at 14.0 min (Figure 5). Therefore, the effects of different tea-making conditions on the acids were investigated (Figure 6). The content of Rosmarinic acid ranged from 3.2 ± 0.2 mg/100 g DW (MD 60 °C) to 526.3 ± 3.8 mg/100 g DW (S5), and that of chicoric acid ranged from 15.2 ± 0.2 mg/100 g DW (MD 80 °C) to 94.7 ± 1.8 mg/100 g DW (S2). The rosmarinic acid content was higher than that of chicoric acid, suggesting that rosmarinic acid was more sensitive to heat treatment (Figure 6). Both rosmarinic and chicoric acid contents were significantly higher in treatments above S2 compared with that in NS and S1. For rosmarinic acid, NS was 58.7 ± 5.9 mg/100 g DW, whereas S2 was 513.1 ± 8.1 mg/100 g DW, which is approximately 8.7 times higher than NS (Figure 6A). In contrast, with respect to chicoric acid content, NS was 74.7 ± 0.2 mg/100 g DW, whereas S2 was 94.7 ± 1.8 mg/100 g DW, which is approximately 1.3 times higher than NS (Figure 6B). The effect of S treatment for more than 2 min was more pronounced for rosmarinic acid content. Both rosmarinic and chicoric acid contents were similar in S treatment of 2–10 min, with some exceptions (Figure 6A,B). For drying methods, FD and MD 40 °C were significantly higher for rosmarinic acid and FD for chicoric acid compared with the other treatments. (*p* < 0.05) (Figure 6a,b). The content of both components was significantly lower in SD compared with that in FD (*p* < 0.05), which was possibly due to drying speed. Drying herbs in the sun or shade may result in the loss of several compounds drawn to the leaf surface, along with the water evaporating during the process. Prolonged air-drying can also result in the enzymatic degradation of components [30]. Both rosmarinic and chicoric acid values were significantly lower at MD 50 °C compared with that at 40 °C and at 60 °C compared with that at 50 °C (Figure 6a,b). Fletcher et al. [31] examined the rosmarinic acid content of spearmint (*Mentha spicata* L), wherein the authors dried the plant at three different temperatures (35, 45, and 80 °C) for 24 to 96 h and via lyophilization at –50 °C, and they reported that it was significantly reduced at 80 °C. Lee et al. [32] also compared chicoric acid content in Echinacea purpurea flowers at three different temperatures (25, 40, and 70 °C) and reported that it decreased with increasing drying temperature. These reports are consistent with the results in this study, wherein the drying temperatures were set to more finely partitioned drying temperature settings (40, 50, 60, 70, 80, 90, and 100 °C) than in the studies by Fletcher et al. [31] and Lee et al. [32]. The results showed that both rosmarinic and chicoric acids decreased drastically (*p* < 0.05) at drying temperatures of 40, 50, and 60 °C and remained constant at 70 °C and above. Therefore, the residual amounts of both rosmarinic and chicoric acids increased with S treatment of 2 min or longer, and drying temperatures of 40 °C or lower are most desirable. This result is consistent with the DPPH and H-ORAC values, suggesting a strong involvement of both components in antioxidant properties. Rosmarinic acid exhibits antioxidant, anti-inflammatory, and anticancer effects [33], and chicoric acid exhibits anti-type 2 diabetes [34], -cancer, -obesity, and -viral properties [32]. In this study, the rosmarinic acid and chicoric acid content could be retained at high levels by drying at ≤40 °C following S treatment for at least 2 min. However, a large peak at approximately 4.0 min retention time in Figure 5B could not be identified. Further investigation of this component is warranted.

### 3.6. Aroma Composition

GC-MS chromatograms of the samples with respect to their treatment are shown in Figure 7. In this study, the number of volatile components detected varied from 42 to 92, depending on the treatment. There were three particularly large peaks (P1, P2, and P3) in Figure 7: P1 was presumed to represent Linalool, a monoterpene present in the aroma components of various fruits, vegetables, spices, coffee, and tea. It is used in cosmetics as a floral perfume, and in confections and soft drinks, because of its lily-like aroma. P2 was presumed to represent *trans*-alpha-bergamotene, which is an aromatic component of Bergamotene with a partially different bonding and atom arrangement. P3 was assumed to be 2-methoxy-3-(2-propenyl)-phenol, a type of phenolic. All three components have been previously reported in basil [35,36]. Figure 8 depicts the change in peak areas (PA) of Linalool, *trans*-alpha-bergamotene, and 2-methoxy-3-(2-propenyl)-phenol with each treatment. During S treatment, the PA of Linalool and 2-methoxy-3-(2-propenyl)-phenol decreased with increasing steam-heating time compared with that during NS, with a remarkable decrease observed > 2 min (Figure 8A,C). The PA of *trans*-alpha-bergamotene was not affected by S treatment, and its duration and showed a remarkable decrease only at S10 (Figure 8B). In some herbs, such as basil, destruction of the glandular trichome and reduction of aromatic compounds have been reported with steam-heat treatment. In this study, SEM images also confirmed the destruction of the glandular trichome at S2 and above (Figure 1A). The glandular trichome is filled with aromatic compounds unique to each plant species. Linalool and 2-methoxy-3-(2-propenyl)-phenol were assumed to have undergone oxidation and hydrolysis reactions [37] and evaporation [38] after the destruction of the glandular trichome, resulting in a decreased PA. On the contrary, *trans*-alpha-bergamotene, which was less affected by S treatment, did not show a decrease in PA even at S5, indicating that it is less prone to chemical reaction and evaporation after glandular trichome destruction compared with other components. The highest PA values for drying methods and temperature were different during SD for Linalool, MD 60 °C for *trans*-alpha-bergamotene, and FD for 2-methoxy-3-(2-propenyl)-phenol. However, a common trend was observed for all three components, with a marked decrease in PA above MD 80 °C and approximately the same value from 80 °C to 100 °C (Figure 8a–c). SEM images showed the destruction of the glandular trichome at S2. However, it remained unchanged, albeit in a deformed state, during MD, even at 100 °C. Since the S treatment involved FD, the heating time was the same as the S treatment time. Therefore, the maximum heating time for S treatment is 10 min. On the contrary, the heating times for MD 80 °C, 90 °C, and 100 °C were 45, 26, and 22 min, respectively, which were longer than those for the S treatment. Therefore, the volatile components of the glandular trichome could have decreased due to the high temperature and prolonged heating time, even without disruption. *Trans*-alpha-bergamotene was less affected by the S treatment than the other two components; however, PA was greatly reduced at S10 (Figure 8B). Therefore, *trans*-alpha-bergamotene is less prone to chemical reactions than Linalool, 2-methoxy-3-(2-propenyl)-phenol but volatilizes or decomposes under high temperature and prolonged heating. Linalool maintained levels similar to those seen during FD of up to MD 70 °C in the absence of glandular trichome disruption. 2-methoxy-3-(2-propenyl)-phenol was the most sensitive to heat among the three components since it decreased at SD and MD 40 °C even without glandular trichome disruption. Therefore, the S treatment should be limited to NS or steaming of 1 min, with respect to retaining aromatic components, whereas FD is the preferred drying method carried out at the right temperature to retain all three components. Therefore, MD ≤ 70 °C should be used for retaining two components (Linalool, *trans*-alpha-bergamotene). 

The analysis of volatile components in this study presents several challenges. The headspace method, which allows for simple analysis, was used, and the values were expressed using only PA without using internal standard reagents. More accurate data could be obtained by extracting volatile components and expressing the ratio of the peak area of volatile compounds to their peak area using an internal standard reagent. Another issue is that the number of measurement repetitions is one. Further investigations of these challenges are warranted.

## 4. Conclusions

In this study, we first determined the antioxidant properties of the leaves, flowers, and stems to identify the appropriate parts of the plant that could be used to produce Genova tea with superior antioxidant properties and quality. The results showed higher levels of antioxidant activity in the leaves and flowers; hence, leaves with the highest yield were used in the tea production experiment. 

Fourteen treatment conditions were set up and examined in detail, focusing on the S times (0, 1, 2, 3, 5, and 10 min), drying methods, and temperatures (–56, 25, 40, 50, 60, 70, 80, 90, and 100 °C). Our results showed that TPC, DPPH, H-ORAC, and rosmarinic and chicoric acid content values were significantly higher after 2 min of steam-heat treatment compared with that NS (*p* < 0.05), especially for the rosmarinic acid. The green color faded after steam-heat treatment and drying at temperatures > 50 °C. The primary aroma components of Genova were Linalool, *trans*-alpha-bergamotene, and 2-methoxy-3-(2-propenyl)-phenol, determined through GC-MS analysis. Among these, Linalool and 2-methoxy-3-(2-propenyl)-phenol were significantly reduced by S treatment, while *trans*-alpha-bergamotene was not. In summary, steam-heating treatment for 2 min is better for retaining high antioxidant properties, drying at <50 °C is better for retaining green color, and the aroma components can be better retained with the absence of S treatment. Materials with high antioxidant properties can be used in the food and pharmaceutical industries, whereas materials with aromatic components can be used in cosmetics.

This study was conducted with the aim of tea production; however, our findings also indicate the possibility of producing an improved Genova dried product, which can be applied in food products, such as tea, as well as in cosmetic and pharmaceutical applications. This study measured only TPC and rosmarinic and chicoric acids as antioxidant components. Identifying other polyphenol components and clarifying the effects of S time, drying method, and temperature on such components will provide novel insights into new developments and applications. Finally, the aroma analysis was evaluated using GC-MS; therefore, more biological evaluation, e.g., human sensory testing, should also be considered in future studies.

## Figures and Tables

**Figure 1 foods-12-01663-f001:**
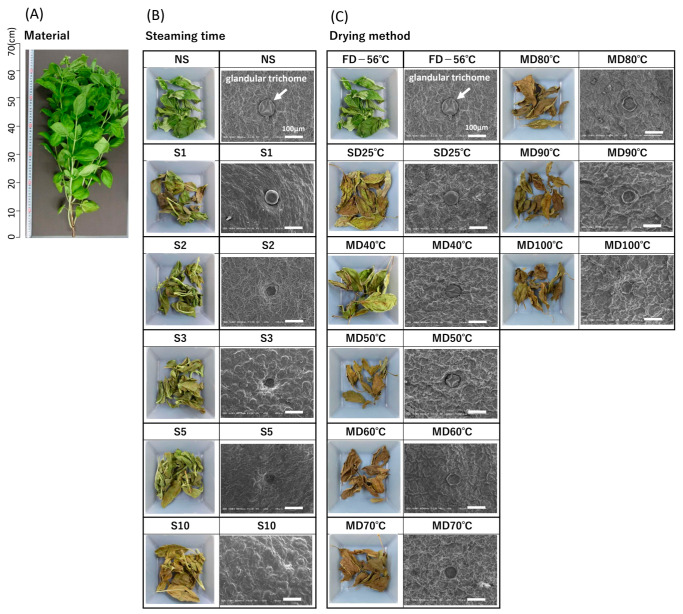
Digital camera and SEM images of *Ocimum basilicum* cv. ‘Genovese’. (**A**) Experimental material, (**B**) dried leaves with different steaming times, and (**C**) drying methods. FD: Freeze-drying (‒56 °C, 48 h), NS: Non-Steaming, S1: Steaming for 1 min, S2: Steaming for 2 min, S3: Steaming for 3 min, S5: Steaming for 5 min, S10: Steaming for 10 min, SD: Shade-drying (25 °C, 7 days), MD 40 °C: Machine-drying at 40 °C, MD 50 °C: Machine-drying at 50 °C, MD 60 °C: Machine-drying at 60 °C, MD 70 °C: Machine-drying at 70 °C, MD 80 °C: Machine-drying at 80 °C, MD 90 °C: Machine-drying at 90 °C, and MD 100 °C: Machine-drying at 100 °C.

**Figure 2 foods-12-01663-f002:**
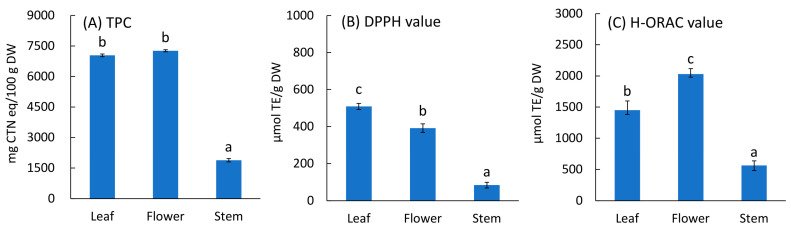
TPC, DPPH, and H-ORAC values of parts of *Ocimum basilicum* cv. ‘Genovese’. TPC: Total polyphenol content, DPPH value: DPPH radical scavenging activity value, H-ORAC value: Hydrophilic oxygen radical absorbance capacity value. The values are expressed as the standard reagents equivalent per dry weight. The results were obtained using Tukey’s test for multiple comparisons. Different letters indicate significant differences at *p* < 0.05. Data are expressed as mean ± SE (n = 6).

**Figure 3 foods-12-01663-f003:**
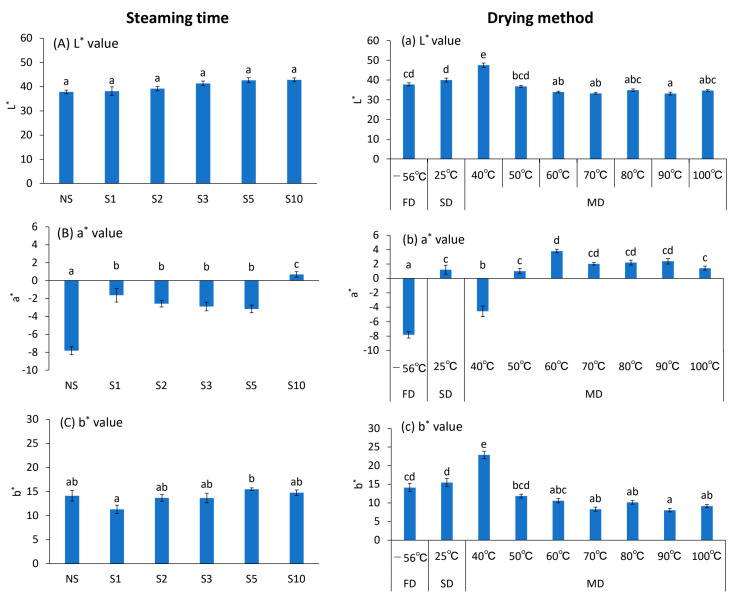
Effects of different treatment conditions on the color of *Ocimum basilicum* cv. ‘Genovese’ tea. (**A**,**a**) L* values, (**B**,**b**) a* values, and (**C**,**c**) b* values. FD: Freeze-drying (‒56 °C, 48 h), NS: Non-Steaming, S1: Steaming for 1 min, S2: Steaming for 2 min, S3: Steaming for 3 min, S5: Steaming for 5 min, S10: Steaming for 10 min, SD: Shade-drying (25 °C, 7 days), MD 40 °C: Machine-drying at 40 °C, MD 50 °C: Machine-drying at 50 °C, MD 60 °C: Machine-drying at 60 °C, MD 70 °C: Machine-drying at 70 °C, MD 80 °C: Machine-drying at 80 °C, MD 90 °C: Machine-drying at 90 °C, and MD 100 °C: Machine-drying at 100 °C. All results were obtained by Tukey’s test for multiple comparisons. Different letters indicate significant differences at *p* < 0.05. Data are expressed as mean ± SE (n = 6).

**Figure 4 foods-12-01663-f004:**
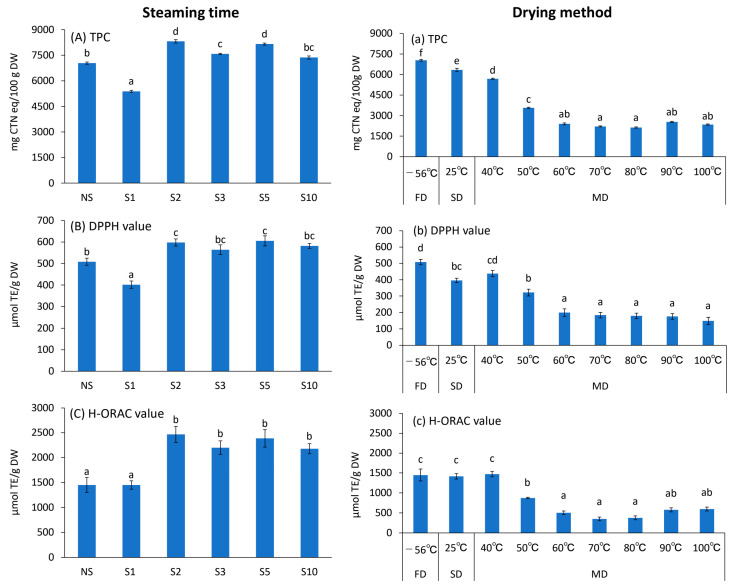
Effects of different treatment conditions on (**A**,**a**) TPC, (**B**,**b**) DPPH, and (**C**,**c**) H-ORAC values of *Ocimum basilicum* cv. ‘Genovese’ tea. FD: Freeze-drying (‒56 °C, 48 h), NS: Non-Steaming, S1: Steaming for 1 min, S2: Steaming for 2 min, S3: Steaming for 3 min, S5: Steaming for 5 min, S10: Steaming for 10 min, SD: Shade-drying (25 °C, 7 days), MD 40 °C: Machine-drying at 40 °C, MD 50 °C: Machine-drying at 50 °C, MD 60 °C: Machine-drying at 60 °C, MD 70 °C: Machine-drying at 70 °C, MD 80 °C: Machine-drying at 80 °C, MD 90 °C: Machine-drying at 90 °C, and MD 100 °C: Machine-drying at 100 °C. The values are expressed as the standard reagents equivalent per dry weight. The results were obtained using Tukey’s test for multiple comparisons. Different letters indicate significant differences at *p* < 0.05. Data are expressed as mean ± SE (n = 6).

**Figure 5 foods-12-01663-f005:**
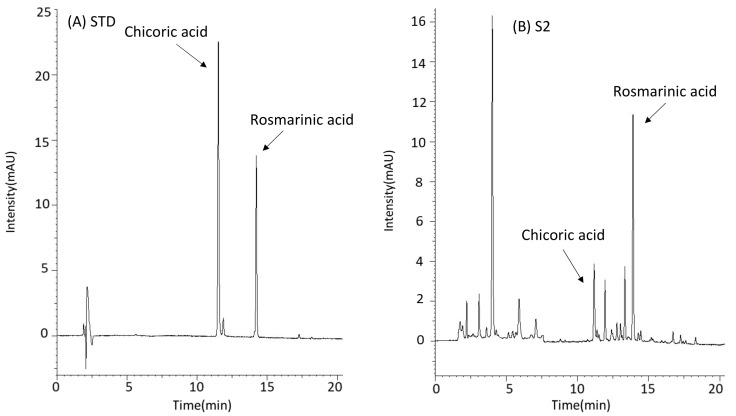
HPLC chromatograms of (**A**) rosmarinic and chicoric acids reagent and (**B**) S2: Steaming for 2 min.

**Figure 6 foods-12-01663-f006:**
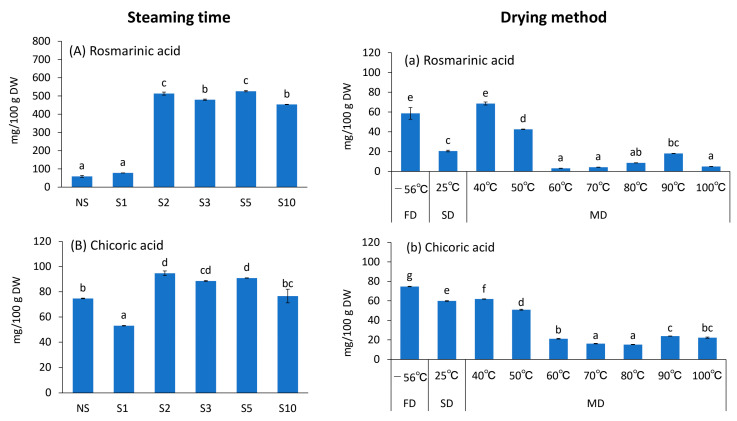
Effects of different treatment conditions on (**A**,**a**) rosmarinic and (**B**,**b**) chicoric acid contents of *Ocimum basilicum* cv. ‘Genovese’ tea. FD: Freeze-drying (‒56 °C, 48 h), NS: Non-Steaming, S1: Steaming for 1 min, S2: Steaming for 2 min, S3: Steaming for 3 min, S5: Steaming for 5 min, S10: Steaming for 10 min, SD: Shade-drying (25 °C, 7 days), MD 40 °C: Machine-drying at 40 °C, MD 50 °C: Machine-drying at 50 °C, MD 60 °C: Machine-drying at 60 °C, MD 70 °C: Machine-drying at 70 °C, MD 80 °C: Machine-drying at 80 °C, MD 90 °C: Machine-drying at 90 °C, and MD 100 °C: Machine-drying at 100 °C. The values are expressed as the standard reagents equivalent per dry weight. The results were obtained using Tukey’s test for multiple comparisons. Different letters indicate significant differences at *p* < 0.05. Data are expressed as mean ± SE (n = 4).

**Figure 7 foods-12-01663-f007:**
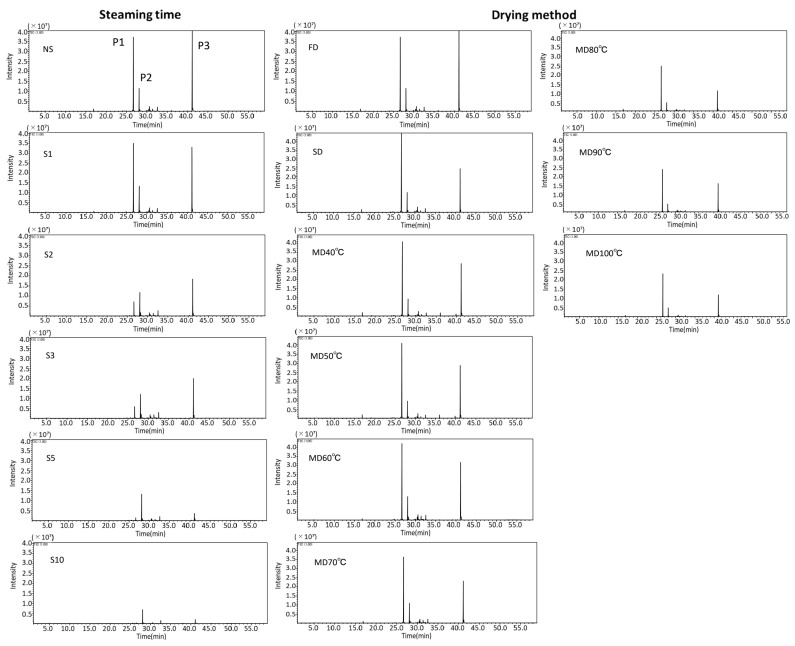
GC-MS chromatogram of *Ocimum basilicum* cv. ‘Genovese’ tea under different tea-making treatment conditions. FD: Freeze-drying (‒56 °C, 48 h), NS: Non-Steaming, S1: Steaming for 1 min, S2: Steaming for 2 min, S3: Steaming for 3 min, S5: Steaming for 5 min, S10: Steaming for 10 min, SD: Shade-drying (25 °C, 7 days), MD 40 °C: Machine-drying at 40 °C, MD 50 °C: Machine-drying at 50 °C, MD 60 °C: Machine-drying at 60 °C, MD 70 °C: Machine-drying at 70 °C, MD 80 °C: Machine-drying at 80 °C, MD 90 °C: Machine-drying at 90 °C, and MD 100 °C: Machine-drying at 100 °C.

**Figure 8 foods-12-01663-f008:**
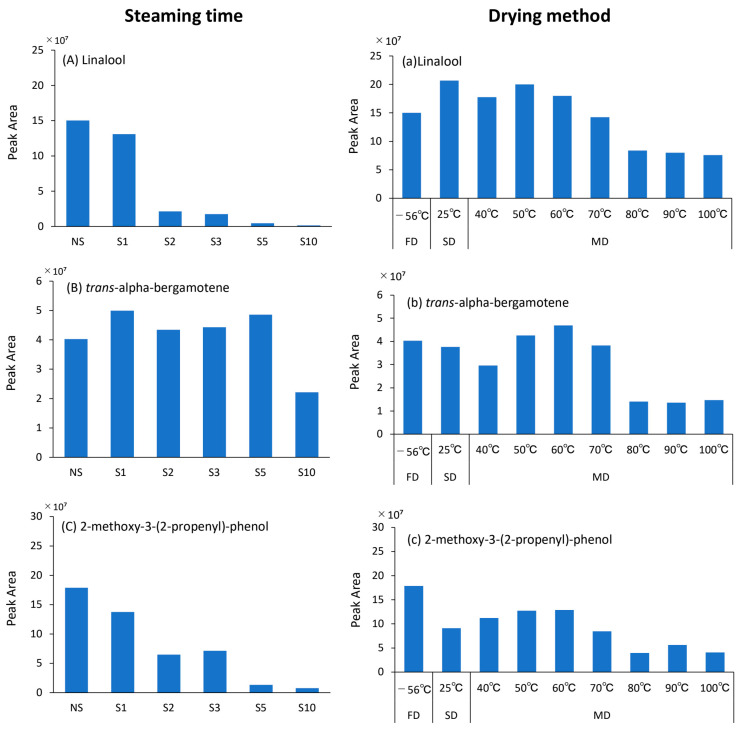
Effects of different treatment conditions on peak areas of aroma compounds in *Ocimum basilicum* cv. ‘Geno-vese’ tea. (**A**,**a**) Linalool, (**B**,**b**) trans-alpha-bergamotene, and (**C**,**c**) 2-methoxy-3-(2-propenyl)-phenol. FD: Freeze-drying (‒56 °C, 48 h), NS: Non-Steaming, S1: Steaming for 1 min, S2: Steaming for 2 min, S3: Steaming for 3 min, S5: Steaming for 5 min, S10: Steaming for 10 min, SD: Shade-drying (25 °C, 7 days), MD 40 °C: Machine-drying at 40 °C, MD 50 °C: Machine-drying at 50 °C, MD 60 °C: Machine-drying at 60 °C, MD 70 °C: Machine-drying at 70 °C, MD 80 °C: Machine-drying at 80 °C, MD 90 °C: Machine-drying at 90 °C, and MD 100 °C: Machine-drying at 100 °C.

## Data Availability

Data is contained within the article.

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
