# Peer review of "Effects of Steam Treatment Time and Drying Temperature on Properties of Sweet Basil’s Antioxidants, Aroma Compounds, Color, and Tissue Structure"

_foods, 2023, doi:10.3390/foods12081663_

Round 1

Reviewer 1 Report

This article analyzed the key quality components in Sweet Basil produced under different processing methods (Steam treatment time and drying temperature); it explored the changes of antioxidant activity, color tone, and aromatic substances in sweet basil under different processes. On the whole, the article is not innovative enough, and the article as a whole lacks logic. The content is relatively thin and the graphical format is relatively single. It needs to be further improved. And major revision is recommended.

1.     AbstractThe description of the results should be more detailed.

2.     Introduction: The study background is lengthy and the significance of the study is not clear. Further improvements are warranted.

3.     Materials: The methodology needs further refinement. Aroma experiments lack biological parallelism. It is also necessary for substances such as caffeic acid. Whether the weight of the leaves is consistent when taking pictures; and the uniformity of the photo environment.

4.     Results and discussion: The logic of the conclusion is not clear and needs further improvement.

5.     Figure 3 and 4, in the drying methods conditions, "A" is recommended to be replaced by "a".

Author Response

Response to Reviewer 1 Comments
We are thankful for the time and effort spent by the Reviewers and Editor in providing
suggestions and comments on our manuscript, which have significantly helped us to improve it.
We have revised the manuscript accordingly (revised text is highlighted in yellow for your
convenience) and hope that our revised manuscript meets your standards and be reconsidered for publication in your esteemed journal. We have provided our point‐by‐point response to all comments raised below.
1. Abstract:The description of the results should be more detailed.
Response

Thank you for your suggestion. We have revised the Abstract accordingly. Please note that due to the strict word limit of the abstract (200 words), there is a limitation to what we can write in the abstract.
2. Introduction: The study background is lengthy and the significance of the study is not clear.Further improvements are warranted.
Response
Thank you for your suggestion. We have revised the introduction accordingly and stated the significance of the study (Lines 51 ‐56).

3. Materials: The methodology needs further refinement. Aroma experiments lack biological parallelism. It is also necessary for substances such as caffeic acid. Whether the weight of the leaves is consistent when taking pictures; and the uniformity of the photo environment.
Response
Thank you for your suggestion. Detailed analysis was added to the Aroma experiments in the Methods section(Lines, 140‐152). The limitations of this study on Aroma experiments were also added (Lines 391‐397, and 449‐451).
In this study, the HPLC analysis was limited to the rosmarinic and chicoric acids based on previous reports.
We agree that analysis on other components, such as caffeic acid, is warranted; therefore, we have added it as a future direction (Lines, 327‐330).
Although the environment at the time of photography was identical, the weights of the tea leaves were not. We have indicated this in the revised manuscript (Lines, 77‐79).

4. Results and discussion: The logic of the conclusion is not clear and needs further
improvement.
Response
Thank you for your suggestion. We have revised the conclusion accordingly (Lines, 438‐451).

5. Figure 3 and 4, in the drying methods conditions, ʺAʺ is recommended to be replaced by ʺaʺ.
Response
Thank you for your suggestion. Accordingly, we have replaced ʺAʺ with ʺaʺ in Figures 3 and 4 and in the drying method conditions. Also, A) was changed to a) for the graph of the drying method in Figure 6 and 8 for consistency. The text has also been revised accordingly.
We have revised the English language expression throughout the manuscript to improve readability.

Reviewer 2 Report

In this work, the effect of tea processing (steaming duration and drying temperature) on the total phenolic content, antioxidant activity, color and aroma compounds of Genova tea were evaluated.

The paper is organized and presented, but the article must be improved. I would suggest the authors make some changes, improvements and suggestions in their manuscript:

Line 12: Antoxidant activity is determined, not measured.

Line 21: “trans” in italics.

Lines 21, 309, 313, etc.: bergamotene in lower case. Really it is: trans-alpha-bergamotene (trans in italics, y alpha without dots before and after the word).

Line 21: 2-methoxy-3-(2-propenyl): The name of this compound is incorrect.

Line 42: sexterpenes ??? The name of this family of compound is wrong.

Section 2.3: How many samples for each treatment were prepared? That is an essential information to see the reproducibility of the obtained results.

Line 84: “…for S-treatment, and Freeze-drying (FD) was done…”

Line 87: “…and 100°C using Machine-drying (MD). All treatments…”

Line 91: “…and Shade-drying (SD) (25°C, 7 days). After tea…”

Figure 1 should be placed after section 2.3.

Lines 100, 167, 172, 198….: Only color (not color tone).

Section 2.5.: How many replicates were performed for color measurements?

Section 2.6: How many extractions (tea) were made to analyze?

Section 2.7.: How many replicates were performed for TPC, and antioxidant measurements?

Line 115: “…is mg catechin equivalent/100 g dry weight (mg CTN eq/100 g DW),…”

Line 118: HPLC methods are quantitative. The chromatographic conditions should be briefly described.

Section 2.8.: How many extractions and injections were performed for liquid-chromatographic analysis?

Line 121:  The extraction of volatile compounds and the chromatographic conditions should be briefly described.

Section 2.9.: How many extractions and injections were performed for gas-chromatographic analysis?

Line 128: “p < 0.05”. Only the “p” in italics. In all manuscript.

Line 168 and following lines: Figure, not photograph.

Figures 5 and 7 are not essential; as there are many figures in the article, they can be moved to a file with supplementary material.

Line 310: 2-methoxy-3-(2-propenyl)-phenol (not phenol, 2-methoxy-3-(2-propenyl)-,).

Line 311: aromatic component found in phenolics ???

Lines 322-322: Phenol, and 2-methoxy-322 3-(2-propenyl)- were assumed… This sentence is wrong.

Section 3.6: Were only those 3 volatile compounds identified? The discussion of their hypothetical content has been done only on the basis of the area of their chromatographic peaks? Has the internal standard method, which is mandatory in gas-chromatographic analysis, been used? Figure 8 should represent the ratio of the peak area of the volatile compound to the peak area of an internal standard.

Author Response

Response to Reviewer 2 Comments

We are thankful for the time and effort spent by the Reviewers and Editor in providing

suggestions and comments on our manuscript, which have significantly helped us to improve it.

We have revised the manuscript accordingly (revised text is highlighted in yellow for your convenience) and hope that our revised manuscript meets your standards and be reconsidered for publication in your esteemed journal. We have provided our point‐by‐point response to all comments raised below.

1.Line 12: Antoxidant activity is determined, not measured.

Response

We have revised the text accordingly (Line 12).

  1. Line 21: “trans” in italics.

Response

We have revised the text accordingly and doubled check the entire manuscript for similar errors.

3.Lines 21, 309, 313, etc.: bergamotene in lower case. Really it is: trans‐alpha‐bergamotene (trans

in italics, y alpha without dots before and after the word).

Response

Thank you for your suggestion. We have revised the text accordingly and doubled check the entire manuscript

for similar errors.

4.Line 21: 2‐methoxy‐3‐(2‐propenyl): The name of this compound is incorrect.

Thank you for your suggestion. We have corrected all references to ʺphenol, 2‐methoxy‐3‐(2‐propenyl)‐ʺ to ʺ2‐methoxy‐3‐(2‐propenyl)‐phenolʺ.

  1. Line 42: sexterpenes ??? The name of this family of compound is wrong.

Thank you for your suggestion. We have corrected it to ʺSesquiterpenesʺ (Line, 39).

  1. Section 2.3: How many samples for each treatment were prepared? That is an essential

information to see the reproducibility of the obtained results.

・Line 84: “…for S‐treatment, and Freeze‐drying (FD) was done…”

・Line 87: “…and 100°C using Machine‐drying (MD). All treatments…”

・Line 91: “…and Shade‐drying (SD) (25°C, 7 days). After tea…”

Response

Thank you for your questions. Each was produced in three iterations. To homogenize the samples, we combined the dried leaves that were produced three times into one. This resulted in one sample per treatment. We have indicated this in the revised manuscript (Lines, 89‐90).

  1. Figure 1 should be placed after section 2.3.

Response

Thank you for your suggestion. We have moved Figure 1 to after Section 2.3.

  1. Lines 100, 167, 172, 198….: Only color (not color tone).

Thank you for your comment. We have corrected all ʺcolor toneʺ to ʺcolorʺ.

9.Section 2.5.: How many replicates were performed for color measurements?

Thank you for your questions. The color measurements were conducted six times per treatment. This was indicated in the revised manuscript (Line 111).

10Section 2.6: How many extractions (tea) were made to analyze?

Thank you for your questions. Extraction was conducted twice per treatment area. This was indicated in the revised manuscript (Lines 117‐118).

11Section 2.7.: How many replicates were performed for TPC, and antioxidant measurements?

Thank you for your questions. TPC and antioxidants were measured in triplicate for one extract. This was

indicated in the revised manuscript (Lines 123‐124).

  1. Line 115: “…is mg catechin equivalent/100 g dry weight (mg CTN eq/100 g DW),…”

Thank you for your suggestion. We have revised the text accordingly (Line 122).

  1. Line 118: HPLC methods are quantitative. The chromatographic conditions should be briefly

described

Thank you for your suggestion. To avoid self‐plagiarism, we only quoted the article. We have added the above information in the revised manuscript (Lines 127‐136).

3

  1. Section 2.8.: How many extractions and injections were performed for liquidchromatographic

analysis?

For HPLC analysis, two hot water extractions were performed, and two injections were made on the same extract, for a total of four replicates (Lines 134‐135).

  1. Line 121: The extraction of volatile compounds and the chromatographic conditions should

be briefly described.

Thank you for your suggestion. To avoid self‐plagiarism, we only quoted the article. As you indicated, we have added the extraction and chromatographic conditions for volatile compounds to the revised manuscript (Lines

140‐150).

  1. Section 2.9.: How many extractions and injections were performed for gas‐chromatographic

analysis?

Thank you for your question. Both the extraction and injection were performed once; this was indicated in the revised manuscript. We also indicated that the number of iterations should be increased (Lined 150 and 395‐397).

  1. Line 128: “p < 0.05”. Only the “p” in italics. In all manuscript.

Thank you for your comment. We have revised this accordingly.

  1. Line 168 and following lines: Figure, not photograph.

Thank you for your suggestion. We have revised the text accordingly.

  1. Figures 5 and 7 are not essential; as there are many figures in the article, they can be moved to a file with supplementary material.

Thank you for your suggestion. Please note that we require Figures 5 and 7 to address the other reviewerʹs point. If possible, we wish to keep them.

20Line 310: 2‐methoxy‐3‐(2‐propenyl)‐phenol (not phenol, 2‐methoxy‐3‐(2‐propenyl)‐,).

As you indicated, we have corrected ʺphenol, 2‐methoxy‐3‐(2‐propenyl)‐ʺ to ʺ2‐methoxy‐3‐(2‐propenyl)‐phenolʺ as suggested.

  1. Line 311: aromatic component found in phenolics ???

Thank you for your question, and we apologize for this error. We have corrected the text (Line 351) .

  1. Lines 322‐322: Phenol, and 2‐methoxy‐322 3‐(2‐propenyl)‐ were assumed… This sentence iswrong.

Thank you for your comment. We have corrected the text accordingly (Line 362).

  1. Section 3.6: Were only those 3 volatile compounds identified? The discussion of their hypothetical content has been done only on the basis of the area of their chromatographic peaks?

Has the internal standard method, which is mandatory in gas‐chromatographic analysis, been used? Figure 8 should represent the ratio of the peak area of the volatile compound to the peak area of an internal standard.

The number of volatile components detected varied from 42 to 92, depending on the processing zone. In the revised manuscript, we indicated that there were three PAs that were particularly large in common (Lines, 344‐346) .The data in Figure 8 are obtained only on the basis of the area of chromatographic peaks [16,18].

16Tsurunaga, Y.; Kanou, M.; Ikeura, H.; Makino, M.; Oowatari, Y.; Tsuchiya, I. Effect of different teamanufacturing methods on the antioxidant activity, functional components, and aroma compounds of Ocimumgratissimum. LWT 2022, 114058.

18Tsurunaga, Y.; Takahashi, T.; Nagata, Y. Production of persimmon and mandarin peel pastes and their uses in food. Food Sci Nutr 2021, 9, 1712‐1719.

The headspace method, which allows for simple analysis, was used, and the values were expressed using only PA without using internal standard reagents. More accurate data could be obtained by extracting volatile components and expressing the ratio of the peak area of volatile compounds to their peak area using an internal standard reagent. Another issue is that the number of measurement repetitions is one. Further investigations of these challenges are warranted. We have added this in the revised manuscript (Lines, 391‐397).

We have revised the English language expression throughout the manuscript to improve readability.